Pepper root rot resistance and pepper yield are enhanced through biological agent G15 soil amelioration

Zhang Xuejiang 1 2 3
Yu Dazhao dazhaoyu1956@126.com dazhaoyu195611@sina.com 1 2 3
Wang Hua wanghua4@163.com 1 2 3
1 Institute of Plant Protection and Soil & Fertilizer, Hubei Academy of Agricultural Sciences , Wuhan , Hubei Provience , China
2 Hubei Key Laboratory of Crop Disease, Insect Pests and Weeds Control , Wuhan , Hubei Province , P. R. China
3 Key Laboratory of Integrated Pest Management on Crops in Central China, Ministry of Agriculture , Wuhan , Hubei Province , P. R. China
Basile Adriana
Electronic publication date: 2021 Jul 19
Publication date: 2021
Volume: 9
Electronic Location ID: e11768
Received 2020 Nov 18; Accepted 2021 Jun 22
Copyright: ©2021 Zhang et al.
Copyright year: 2021
Copyright holder: Zhang et al.
License: This is an open access article distributed under the terms of the Creative Commons Attribution License, which permits unrestricted use, distribution, reproduction and adaptation in any medium and for any purpose provided that it is properly attributed. For attribution, the original author(s), title, publication source (PeerJ) and either DOI or URL of the article must be cited.
License URL: https://creativecommons.org/licenses/by/4.0/

Keywords: Capsicum annum L. Root Rot, Biological Agent G15, Microbial Community Structure

Funding: The Key Research and Development Project No. 2017YFD0201600 The National Key Research and Development Program of China No. 2017YFD0200605 This work was financially supported by the Key Research and Development Project (No. 2017YFD0201600), and the National Key Research and Development Program of China (No. 2017YFD0200605). The funders had no role in study design, data collection and analysis, decision to publish, or preparation of the manuscript.

==============================
Pepper root rot is a serious soil-borne disease that hinders pepper production, and efforts are being made to identify biological agents that can prevent and control pepper root rot. Our group recently discovered and produced a biological agent, named G15, which reduces the diversity and richness of fungi and bacteria when applied to pepper fields. In the soil of the G15-treatment condition, the pathogenic fungus Fusarium was inhibited, while the richness of beneficial bacteria Rhodanobacter was increased. Also, the ammonia nitrogen level was decreased in the G15-treatment soil, and the pH, total carbon, and total potassium levels were increased. Compared to the control condition, pepper yield was increased in the treatment group (by 16,680 kg acre−1). We found that G15 could alter the microbial community structure of the pepper rhizosphere. These changes alter the physical and chemical properties of the soil and, ultimately, improve resistance to pepper root rot and increase pepper yield.

Introduction

Pepper (Capsicum annum L.) root rot is a serious soil-borne disease caused mainly by Fusarium sp. and Phytophthora capsici Leon (Aboelnaga & Ahmed, 2007; Pérez Hernández et al., 2014). The Hymenophora subfamily is responsible for pepper root rot, and the family members include F. solani, F. vasinfectum Atk., F. equiseti (Corda) Sacc., F. moniliforme, F. oxysporum Schlecht., and F. verticillioide (Jaber & Alananbeh, 2018). The pathogen invades the plant vascular bundle from the micro-wounds on the roots and stems of the pepper, causing mycelium, sclerotia, and chlamydospores to overwinter in the plant residues and field soil. The chlamydospores of the pathogen are highly resistant to stress, and they generally live in the soil for 3 to 4 years and even up to 10 years (Beckman, 1987). These conidia are spread by irrigation (e.g., water, rainwater, and dew). The development of intensive cultivation and facility cultivation technologies, the use of long-term continuous cropping and simple rotation in vegetable fields has led to the continuous decline of soil quality, where the number of beneficial microorganisms in the soil has plummeted alongside the accumulation of pathogens (Hussain et al., 2009). Therefore, peppers grown in this soil are more prone to root rot (Ma et al., 2008; Ikeda, 2010; Coolon et al., 2013; Cao et al., 2014; Sun et al., 2015).

The control methods of pepper root rot mainly include agricultural control (reasonable rotation, especially film-covered high-ridge cultivation, and seed disinfection), chemical control (40% fluosilazole emulsifiable concentrate, 25% propiconazole emulsifiable concentrate, 3% Guangkuling + 70% metoprolol, dibenzofuran, and diisooctyl phthalate), biological control (Bacillus amylolique Faciens Ohkuma et al., 2001; Chen et al., 2007), B. subtilis wettability Powder (Jayaraj et al., 2010), and breeding and screening for disease-resistant varieties. The biological control of crop root rot is mainly Trichoderma, such as T. harzianum, T. viride, T. hamatum, and T. longibrachiatum, T. polysporum and T. asperellum, etc. The trichomycin secreted by these fungi is the main substance that inhibits crop root rot. A variety of Bacillus species are also important biocontrol bacteria that inhibit root rot of crops. The antibacterial substances produced by them mainly dissolve cell walls or cell membranes, causing protoplasm leakage to break or deform the hyphae, and inhibit the germination of pathogenic bacteria spores to achieve the effect of inhibiting pathogenic bacteria (Ohkuma et al., 2001). Existing biological agents, such as B. subtilis, B. cereus, and B. amyloliquefaciens, change the composition of the soil microbial community in the rhizosphere (e.g., by increasing bacterial diversity or reducing fungal diversity), and increasing pepper resistance to root rot (Francis, Holsters & Vereecke, 2010; Bhat, 2013; Han et al., 2019). Alternatively, introducing certain biological agents might be effective, to some extent, as bio-fertilizers and, at the same time, offer different mechanisms for controlling plant disease rather than chemical pesticides (Fravel, 2005; Mehta et al., 2014; Luo et al., 2018; Ekea et al., 2019).

Therefore, maintaining a high level of microbial diversity in the soil is essential for the sustainable development of the pepper industry (Kennedy & Smith, 1995; Bhat, 2013). It has been reported that inorganic fertilizers can reduce the diversity and abundance of bacteria (Ramirez et al., 2010; Coolon et al., 2013; Zhou et al., 2015), but the application of organic fertilizers and fungal inocula supports the development of soil microbial communities, with greater biodiversity in long-term fertilized soils (Ding et al., 2016).

The microbial fertilizer (G15) developed by our laboratory, which uses Bjerkandera sp. as the main microbial source (Wang, 2018 patent number: ZL 2015 1 0564715.4; Wang et al., 2020 patent number: 2020101644), has a good inhibitory effect on a variety of soil-borne bacteria and fungal diseases. The G15 is fermented by Bjerkandera sp., pig manure, straw, soybean meal, urea, etc. in proportion, and is highly safe to crops. From the perspective of plant growth, the plant height, root length, whole plant fresh weight, and root fresh weight of the G15-treatment pepper were significantly different when compared to the control condition (Cai et al., 2019). Therefore, the current study aims at assessing the microbial variation of the pepper rhizosphere, evaluating the effects of G15 on controlling the pepper root rot disease and, at the same time, improving the nutritional status of the grown plants, thus maintaining pepper field sustainability.

Materials & Methods

Site description and experiment layout

Field experiments were approved by the Research Council of the Institute of Plant Protection and Soil & Fertilizer, Hubei Academy of Agricultural Sciences (project number: 18.010.15).

The Institute of Plant Protection and Soil & Fertilizer, Hubei Academy of Agricultural Sciences granted Ethical approval to carry out the study within its facilities (Ethical Application Ref: hb358-a7c6d).

The field experiment was located in Xinhua town of Shennongjia Forest, Hubei province, China (31°59′N, 110°89′E). This region has a subtropical monsoon climate, with an average annual temperature of 12 °C and 1170.2 mm of precipitation. The land belongs to flat terrain and uniform soil quality. The soil in this field was characterized as mountain dark brown soil with a pH of 5.35 (10:1 water to soil ratio), and it contains 0.783 g kg−1 total N, 4.51 g kg−1 total P, and 46.31 g kg−1 total K (Liang et al., 2011). The field experiment was performed in a completely randomized block design with three replicates for each of the two treatments: G15 treatment and the control condition. which can ensure that the initial conditions of C and G are exactly the same. The cultivation and management methods of the control and treatment are the same, except that G15 microbial fertilizer is not added. G15 is a kind of microbial fertilizer, pig manure, straw, corn stover, sugar, urea, and water, which are necessary as its substrate. This is also the biggest difference between microbial fertilizer and microbial inoculants. The area of each repeat is 300 m2. There are 1,000 peppers in 300 m2. Ridge forming first, 0.8 m ridge distance, the powdery G15 microbial fertilizer was applied to the middle of the ridge distance at one time. The ridge forming conditions of the control were consistent with G15 treatment. The tested pepper variety is “Xiangshuai”. The bred pepper seedlings were planted on May 21, 2019, with plant spacing 0.4 m. Cultivation and water management are carried out according to conventional cultivation management mode.

Field pepper biological character and disease index survey

When the pepper is grown about two months, that is, in mid-July, the physiological characteristics of the plant, including the plant height, root length, root fresh weight, and root dry weight were measured. These indicators can reflect the growth-promoting effect of G15 microbial fertilizer on pepper. The stronger the plant, the stronger the resistance to root rot. We also recorded the incidence of root rot disease and tested whether G15 treatment reduced pepper root rot. Chinese peppers are harvested in five to six batches during the picking period. After the third batch, pepper root rot disease has obvious symptoms. Therefore, collecting pepper rhizosphere soil samples at this time can more truly reflect the changes in soil microorganisms so as to better explain the mechanism of G15 microbial fertilizer to prevent and control pepper root rot. Picking peppers began around July 25, and picking a batch of peppers every 10 to 12 days, a total of five batches were picked. Regarding the harvesting statistics of peppers, the comparison can only be made after selecting three batches. Five batches of peppers treated with G15 were picked. The height of the pepper plant is measured with a meter ruler, and the root length is measured with a centimeter ruler. Both have one decimal place, but when the average value is reached, two decimal places are kept. After all batches were picked, the total production statistics were calculated.

The disease is classified according to the ratio (%) of the diseased root system to the total root system: grade 0, no disease; grade 1, the root system is slightly discolored, and the discolored root system accounts for less than 10% of the total root system, and the plant does not wilt; grade 3, the root system is obviously browned , the discolored root system accounts for 10–30% of the total root system, and the plant begins to wilt; at level 5, the discolored root system accounts for 30% to 50% of the total root system, and the plant is obviously wilting; at level 7, the discolored root system accounts for 50% to 80% of the total root system, plants are wilting; level 9, the whole plant is dead (Sun et al., 2015). Sixty plants was measured in each plot, including healthy and diseased plants.

Soil sampling and properties analysis

The focus of research was the role of G15-treatment in the prevention and control of pepper root rot when root rot disease occurred in pepper. The only difference between the control and the treatment was whether or not G15 microbial fertilizer was added, so this experiment did not take the soil sample before treatment.

Soil sampling was performed between the second and third picks of peppers. Due to ground flatness, uneven water and fertilizer, the soil of the same plot will have differences in the microbial community structure. In order to reduce the impact of these differences on the analysis of the sample community structure, a five-point sampling method will be used in the set plot perform sample mixing. Each treatment took five points and mixed them into one sample. There were three replicates in the same treatment group, and a total of three samples were taken.

It is to study the microbiome in environmental samples, so the DNA of the entire soil sample must be extracted. In addition to the soil, the rhizosphere soil samples will also contain some litter and plant roots. If they are not screened, the DNA information of these litters and plant roots will also be extracted together, which will affect the effective data of sequencing. This kind of operation is to reduce the deviation and will be used in many documents. The bulk soil separated from the root was air-dried for the determination of physicochemical properties [pH, total organic carbon (TOC), NH4+-N, total N (TN), total P (TP), total K (TK)] following the methods described in Shen et al. (2013).

DNA extraction, PCR amplification, library preparation, and Miseq sequencing

Microbial DNA was extracted from six samples using the E.Z.N.A.® soil DNA Kit (Omega Bio-Tek, Norcross, GA, USA) according to the manufacturer’s protocols. The final DNA concentration and purity were determined using a NanoDrop 2000 UV-vis spectrophotometer (Thermo Scientific, Wilmington, DE, USA), and the DNA quality was checked by 1% agarose gel electrophoresis. DNA extracted from each soil sample served as a template for the amplification of the 16S rRNA gene and the internal transcribed spacer region. Bacterial primers 338F (5′-ACT CCT ACG GGA GGC AGC AG-3′) and 806R (5′-GGA CTA CHV GGG TWT CTA AT-3′) were used to amplify the V3-V4 hypervariable regions of the bacterial 16S rRNA gene, while the ITS1 region of the fungal ITS was targeted by ITS1F (5′-CTT GGT CAT TTA GAG GAA GTA A-3′) and ITS4 (5′-GCT GCG TTC TTC ATC GAT GC-3′). When doing amplicon sequencing, it is generally not technically repeated on DNA samples, but on PCR products. After the DNA is extracted in this experiment, 3 PCR repeats will be performed on each sample, and the PCR products of these 3 repeats will be mixed for subsequent library construction and sequencing. The resulting PCR products about 300 base pair were extracted from a 2% agarose gel and further purified using the AxyPrep DNA Gel Extraction Kit (Axygen Biosciences, Union City, CA, USA) and quantified using QuantiFluor-ST (Promega, USA) according to the manufacturer’s protocols. Purified amplicons were pooled in equimolar and paired-end sequenced (2 × 300) on an Illumina MiSeq platform (Illumina, San Diego, USA) according to the standard protocols by Majorbio Bio-Pharm Technology Co. Ltd. (Shanghai, China). The raw reads were deposited into the NCBI Sequence Read Archive (SRA) database (Accession Number: SRP265722).

Sequence data processing

Raw FastQ files were demultiplexed, quality-filtered by Trimmomatic, and merged by FLASH using the following criteria: (i) The reads were truncated at any site receiving an average quality score <20 over a 50 bp sliding window; (ii) Primers were exactly matched, allowing two nucleotide mismatching, and reads containing ambiguous bases were removed; (iii) Sequences with an overlap of longer than 10 bp were merged according to their overlap sequence. Operational taxonomic units (OTUs) were clustered with 97% similarity cutoff using UPARSE (version 7.1; http://drive5.com/uparse/), and chimeric sequences were identified and removed using UCHIME. The taxonomy of each 16S rRNA gene sequence was analyzed by the RDP Classifier algorithm (http://rdp.cme.msu.edu/) against the Silva (SSU123) 16S rRNA database using a confidence threshold of 70%.

Statistical analysis

An OTU-based analysis was performed to detect the microbial community richness and diversity between biological agent treatment and control. Richness was estimated using the Chao index, while the Shannon diversity index was calculated to estimate the number of observed OTUs that were present.

All statistical tests performed in this study were considered statistically significant at P < 0.05. The data were tested for normality and transformed when necessary to meet the criteria for a normal distribution. Duncan and pairwise comparison tests were used to assess the effect of G15 treatment on pepper yield and microbial community, respectively. Multiple analysis of variance (MANOVA) using the SigmaPlot software program was used to determine the effects of G15 (relative to the control condition) on the dependent variables, soil characteristics, relative abundances of abundant taxa, and α-diversity indices, including the Chao and Shannon indices.

Differences in microbial community composition when comparing the G15-treatment and control conditions were tested by ANOSIM. Non-metric multidimensional scaling (NMDS) based on the Bray-Curtis distance was performed to illustrate the ß-diversity for bacteria and fungi.

Results

Impacts of G15-treatment on pepper root rot, biological characteristics, pepper yield

G15 application significantly (Duncan test, P < 0.05) increased plant height (by 17.87 cm), root length (by 9.56 cm), root fresh weight (by 9.56 g), root dry weight (by 2.13 g), and pepper yield (by 16,680 kg acre−1) (Table 1). By enhancing the growth of pepper plants, the plants themselves are strong. Compared with the control treatment, the G amendment has a disease index of only 3.22, and the control effect reached 77.5%.

Impacts of G15-treatment on soil physicochemical characteristics

Supplementing the soil with G15 generally resulted in significantly higher soil pH, TOC, and TK, but the concentrations of NH4+-N decreased significantly relative to the control condition (Table 2). No significant differences in soil TN and TP were detected when comparing the G and C conditions, and their interaction terms were also not significant.

Impacts of G15-treatment on microbial community α-diversity

We detected big variation in the estimated richness and diversity indices when comparing the bacterial and fungal communities of the G15 and C soils (Table 3). From the Sobs and Chao indices (reflecting the community richness) and the Shannon and Simpson indices (reflecting the community diversity), the G15 supplemented soil had a significantly lower richness and diversity for bacteria and fungi relative to the control condition.

Table 1 Pepper biological characteristics, disease indices, and control effect.

Treatments	Plant height (cm)	Root length (cm)	Root fresh weight (g)	Root dry weight (g)	Pepper yield	Disease index	Control effect (%)	
					First pick (kg)	Second pick (kg)	Third pick (kg)	Fourth pick (kg)	Fifth pick (kg)			
C1	45.72 ± 3.94	16.73 ± 2.40	12.59 ± 2.48	2.48 ± 0.32	513 ± 15.47	457 ± 7.70	268 ± 7.23	–	–	16.87 ± 2.68	–	
C2	47.64 ± 4.10	18.35 ± 2.33	13.57 ± 1.93	2.75 ± 0.25	527 ± 9.46	468 ± 5.98	273 ± 9.04	–	–	17.86 ± 1.82	–	
C3	43.93 ± 5.16	14.78 ± 2.61	9.05 ± 1.95	2.33 ± 0.23	508 ± 9.86	449 ± 6.26	261 ± 8.82	–	–	15.64 ± 1.23	–	
G1	61.83 ± 1.45	23.45 ± 3.51	18.93 ± 3.98	4.35 ± 0.27	586 ± 17.17	513 ± 9.63	458 ± 8.85	415 ± 5.66	287 ± 8.92	2.35 ± 0.29	76.53 ± 1.58	
G2	63.75 ± 3.63	25.89 ± 3.30	20.37 ± 2.68	4.67 ± 0.35	597 ± 15.17	526 ± 8.08	467 ± 8.26	423 ± 6.80	291 ± 5.85	3.17 ± 0.24	77.69 ± 2.21	
G3	65.31 ± 3.70	27.35 ± 3.62	24.58 ± 2.66	4.93 ± 0.28	618 ± 6.73	531 ± 9.91	477 ± 9.70	431 ± 6.96	293 ± 5.02	4.13 ± 0.38	78.32 ± 1.95	
Notes.

G biological agent treatment

C control condition

Impacts on microbial community structure

Sample hierarchical clustering analysis and non-metric multidimensional scaling (NMDS) were performed in each community for fungi and bacteria, respectively (Fig. 1). Actually, the clustering effect by treatment was the same in the fungi and bacteria analysis.

A heatmap analysis of the genera-level community richness of fungi and bacteria (top 50) found that only two-fifths of the fungal genera had a richness greater than 1, and the community structure was relatively simple. Three-quarters of the genera of bacteria had an richness of more than 1, and the community structure was more complicated (Fig. 2).

Table 2 Selected physicochemical characteristics for soils according to biological agent treatment and control condition.

Items	pH	TOC mg/g	NH4+-N mg/kg	Total N mg/kg	Total P g/kg	Total K g/kg	
G1	7.56 ± 0.04	39.6 ± 2.9	39.24 ± 5.9	763.4 ± 15.9	4.91 ± 0.12	54.21 ± 2.9	
G2	7.48 ± 0.07	36.4 ± 3.8	28.38 ± 5.2	812.6 ± 25.2	4.63 ± 0.21	56.23 ± 3.9	
G3	7.28 ± 0.09	38.2 ± 4.4	39.46 ± 4.4	694.2 ± 26.4	4.32 ± 0.23	52.16 ± 4.7	
C1	5.68 ± 0.08	10.2 ± 2.4	116.26 ± 8.6	685.6 ± 28.6	4.36 ± 0.20	47.23 ± 4.6	
C2	5.24 ± 0.11	9.6 ± 2.1	136.64 ± 5.2	786.4 ± 25.2	4.96 ± 0.24	45.36 ± 5.2	
C3	5.62 ± 0.08	11.6 ± 2.3	128.28 ± 6.4	826.5 ± 16.4	4.82 ± 0.21	45.98 ± 4.4	
Notes.

G biological agent treatment

C control condition

Table 3 Alpha diversity index analysis reflecting community richness, community evenness, community diversity, and community coverage between the G15-treatment and control conditions.

	Fungi	Bacteria	
Estimators	C-Mean	G-Mean	P-value	C-Mean	G-Mean	P-value	
shannon	2.8632 ± 0.1075	2.0811 ± 0.1065	0.0009	4.1546 ± 0.5282	3.13 ± 0.28	0.0412	
simpson	0.0831 ± 0.0099	0.1951 ± 0.0208	0.0011	0.0346 ± 0.0218	0.1748 ± 0.0484	0.0102	
sobs	83 ± 6.9282	53.333 ± 1.1547	0.0019	281 ± 43.589	239 ± 7.8102	0.1758	
ace	89.725 ± 6.9945	62.891 ± 9.1255	0.0156	300.34 ± 38.639	258.17 ± 6.1596	0.1353	
chao	87.958 ± 7.8902	59.917 ± 6.8795	0.0097	302.99 ± 45.157	264.31 ± 11.553	0.224	
coverage	0.9997 ± 0.0001	0.9997 ± 0.0001	0.588	0.999 ± 0.0002	0.999 ± 0.0002	1	
Notes.

G biological agent treatment

C control condition

Figure 1 Comparative analysis of OTUs at genus level in each community.

(A & C) Fungi and (B & D) bacteria. (A and B) Sample hierarchical clustering analysis based on OTU level with distance algorithm based on bray_curtis. (C and D) Sample NMDS analysis based on the genus level. G = biological agent treatment, C = control condition.

Figure 2 Heatmap analysis of genus horizontal community.

(A) Fungus, and (B) bacterial. Only two-fifths of the fungal genera had a richness greater than 1, and the community structure was more complicated. G = biological agent treatment, C = control condition.

Impacts of G15-treatment on microbial community taxonomic composition

Classified sequences across all samples were affiliated with one bacterial phylum and eight fungal phyla.

For the fungal population, there were 39 genera in the two treatments-conditions and 22 endemic genera in control, and there was a special Pezizales in the G15-treatment. The fungi were mostly rotted on humus-rich soil, plant residues, or manure. For the bacterial population, there were 245 genera in total in the two treatments, 120 endemic genera in the control condition, and only 47 endemic genera in the G15- treatment (Fig. 3).

Figure 3 Venn analysis based on the genus level.

(A) Fungi, (B) bacteria). For the fungal population, there were 39 genera in the two treatments and 22 endemic genera in control, and there was a special Pezizales in the G treatment. For the bacterial population, there were 245 genera in total in the two treatments, 120 endemic genera in the control condition, and only 47 endemic genera in the G treatment. G = biological agent treatment, C = control condition.

Fungal diversity analysis found that the proportion of Phialemonium in the G15 treatment group was significantly higher than that of the control group, while the proportion of Fusarium—that is responsible for pepper root rot—was significantly lower than that of the control group. Bacterial diversity analysis found that the proportion of Rhodanobacter in the G15 treatment group was much higher than in the control group (Fig. 4).

Figure 4 Analysis of microbial community composition at the genus or species level.

(A & B) Fungi and (C & D) bacteria. Fungal diversity analysis found that the proportion of Phi al emonium in the G15 treatment group was significantly higher than of the control group, while the proportion of Fusarium –that is responsible for pepper root rot –was significantly lower than that of the control group. Bacterial diversity analysis found that the proportion of Rhodanobacter in the G15 treatment group was much higher than in the control group. G = biological agent treatment, C = control condition.

Discussion

In vegetable fields with long-term application of chemical fertilizers, the soil acidity is generally maintained at around pH 5.35, and in some places the soil acidity is even lower. Excessive acidification of the soil is extremely detrimental to pepper growth, which leads to short, thin plants that are susceptible to root rot disease, resulting in extremely low pepper production (Ekea et al., 2019). The G15 microbial fertilizer greatly changed the physical and chemical properties of pepper rhizosphere soil. The pH value of G15 microbial fertilizer is about 8.4. After several months of growth of pepper, the microbial fertilizer interacts with the soil and the pepper roots, and the pH value of the rhizosphere soil reaches about 7.5. Compared to G15 treatment, the pH value of the control group is around 5.5. The increase in the pH following the application of G15 also contributes to the growth inhibition of pathogenic bacteria and likely has a preventive effect (Cai et al., 2019).

At the same time, we also found that in G15 treatment, the K content of rhizosphere soil was significantly higher than that of the control group. Some studies think the soluble forms of K are good for plant growth and yield (Singh, Maurya & Verma, 2014). The treatment group can accumulate more K ions and more carbon, which is also important for enhancing the nutrient absorption of pepper and resistance to pepper root rot. After applying G15 to the pepper field, we found that the root system of the G15 treated plants was more developed, which helps to resist the invasion of root rot fungi; the stem was thicker and can resist lodging, which is beneficial to nutrient absorption; the leaves were more green, which presumably benefited photosynthesis. We found that the G15 treated plants had a longer flowering time than that the control plants, and the fruit-bearing period was about one month longer in the G15 treated plants than that in the control plants. Pepper production was increased by 16,680 kg acre−1 in the G15-treatment condition relative to the control condition.

The main function of G15 microbial fertilizer is to prevent and suppress diseases, while promoting plant growth and increasing crop yield. Even if G15 microbial fertilizer is applied, some root rot diseases of peppers are normal. Because the disease prevention efficiency reaches more than 70%, it shows that the effect of this microbial fertilizer is very efficient. To explore the disease-resistance mechanism of G15, we studied the microbial diversity and community structure of the pepper rhizosphere. We found that the fungal and bacterial diversity and community richness of the G15 treated soils were lower than those of the control condition, which is somewhat different from the findings of a previous study (Qiao et al., 2019). In the G15-treatment condition, the fungal and bacterial community structures were relatively simpler, but the richness of the genera was higher than those in the control condition. This might be related to the application of G15 and the mode of administration. In our experiment, G15 was applied by stripping rather than spreading. The strip application method concentrates the G15 in the ridge area where the pepper is planted, which greatly changes the micro-environment around the root of the pepper. A unique Pezizales was found from the G15 treatment in the fungal community, which was more abundant. The Pezizales were mostly rotted on humus-rich soil, plant residues, or manure. This is also the result of the fungal effect by concentrating on the root of the pepper.

The difference between microbial fertilizers and microbial inoculants is that the microorganisms can be compounded in the corresponding matrix to better function. However, because there is no matrix, the microbial inoculants are directly applied to the field, and their effects will be affected by many factors, resulting in great effectiveness reduce. In the past experiments of applying organic fertilizer, it can also increase production, and the microbes in the rhizosphere of crops will also change, but the changes of these microbes cannot interact with the root rot of the crops, thereby preventing and controlling crop root rot (Tao et al., 2020). Therefore, the difference in microbial changes in the rhizosphere of pepper caused by the application of G15 microbial fertilizer in this study is the main reason for inhibiting pepper root rot.

The pepper root rot-causing Fusarium genus is much more abundant in the control group than in the G15 treatment group. We propose that after the application of G15, the fungus interacts with pepper roots and the rhizosphere soil. The fungus inhibits the growth of some other fungi and bacteria in the soil, resulting in the reduced fungal abundance, diversity, and richness in the treatment group. The fungus itself has a strong inhibitory effect on Fusarium, resulting in an extremely low abundance of Fusarium in the treatment group.

At the same time, the application of G15 increased the richness of some beneficial bacteria (e.g., the richness of Rhodanobacter sp was much higher in the G15-treatment condition than in the control condition). Rhodanobacter sp is a denitrifying bacterium that can efficiently degrade ammonia nitrogen in the soil (Prakash et al., 2012; Green et al., 2012; Hemmc et al., 2016). However, there is relatively few research on its biological control effect. Dománski (1982) isolated from the bark of American red oak and scutellaria, a strain of Fumaria tuberculosis that inhibits Glycophyllum vulgaris, effectively preventing brown rot from occurring on oak trees. Bak et al. (2011) isolated a strain of biocontrol fumigatus in a planting area contaminated by pathogenic bacteria in Korea. The results of the study showed that fumigatus could grow on PDA medium at 5 °C, 35 °C, and the bacteria could also significantly inhibit the pathogen Lentinula edoes, effectively prevent the occurrence of dry rot on black poplar in Europe and America. Wang, Yu & Guo (2015) disclosed that tobacco tube fungus has a good inhibitory effect on root rot, verticillium wilt, bacterial wilt, etc., and found that the granules prepared with this strain can significantly control tomato bacterial wilt, blight, cotton root rot, verticillium wilt, watermelon root rot wilt, etc. Rhodanobacter strains have different denitrification capabilities at the genetic level (Hemmc et al., 2016). In an acidic and nitrate-rich environment, Rhodanobacter strains exhibit high relative abundance and activity, and all have complete denitrification capabilities (Green et al., 2012), indicating that Rhodanobacter strains may maintain their own growth using high concentrations of the nitrates. During the growth process of pepper, if the ridge is not sufficiently high and is waterlogged, pepper root rot is more likely happened. Therefore, in the G15-treatment condition, the extremely rich Rhodanobacter might efficiently degrade excessive ammonia nitrogen in the pepper rhizosphere, thereby reducing the occurrence of pepper root rot. This can also be seen from the soil’s physical and chemical indicators. The ammonia nitrogen in the treatment group was only a quarter of that in the control group, indicating that the beneficial bacteria Rhodanobacter sp in the treatment group plays an important role in degrading excess ammonia nitrogen in the rhizosphere. This might be the key to the G15-mediated inhibition of pepper root rot.

Conclusions

It is notoriously difficult to identify biological agents that contribute to pepper root rot resistance and increase pepper yield. Nevertheless, our group identified the biological agent G15, which improves the richness of the beneficial bacteria Rhodanobacter sp. and decreases the abundance of the pathogenic Fusarium genus. G15 supplementation alters the physical and chemical properties of the soil in a way, especially increasing the pH value of the soil, and enriching K, that improves the nutrition utilization of pepper, enhances pepper root rot resistance, and increases pepper yield.

Additional Information and Declarations

Competing Interests

Author Contributions

Ethics

Field Study Permissions

Patent Disclosures

Data Availability

The authors declare there are no competing interests.

Xuejiang Zhang and Hua Wang conceived and designed the experiments, performed the experiments, analyzed the data, prepared figures and/or tables, authored or reviewed drafts of the paper, and approved the final draft.

Dazhao Yu conceived and designed the experiments, performed the experiments, analyzed the data, prepared figures and/or tables, and approved the final draft.

The following information was supplied relating to ethical approvals (i.e., approving body and any reference numbers):

The Institute of Plant Protection and Soil & Fertilizer, Hubei Academy of Agricultural Sciences granted Ethical approval to carry out the study within its facilities (Ethical Application Ref: hb358-a7c6d).

The following information was supplied relating to field study approvals (i.e., approving body and any reference numbers):

Field experiments were approved by the Research Council of the Institute of Plant Protection and Soil & Fertilizer, Hubei Academy of Agricultural Sciences (project number: 18.010.15).

The following patent dependencies were disclosed by the authors:

Wang H. 2018. Bjerkandera sp. Gao’s No. 15 and its preparation for preventing and curing vegetable root diseases. CHN. Patent ZL 2015 1 0564715.4. 2018-01-02.

Wang H., Wang HY, Liu CP, Zhang Y. Zhang XJ, Liu YM. 2020. A fermentation material and its preparation method used for sprout cultivation substrate of solanaceous vegetables. Australia, Patent number: 2020101644.

The following information was supplied regarding data availability:

The data are available at NCBI: SRP265722.

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
