# Peer review of "Pepper root rot resistance and pepper yield are enhanced through biological agent G15 soil amelioration"

_PeerJ, doi:10.7717/peerj.11768_

## Round 0.1 · original submission · Major Revisions

I recommend that the authors follow all the indications of the reviewers and respond adequately and completely to the requests of the reviewer 2.

·

Basic reporting

The article was written in clear, unambiguous, technically correct English and conforms to professional standards of courtesy and expression. Neverthelee I suggest some small improvements, which are listed in the attached file, in order to improve grammar and some scientific expressions. For example, throughout the paper the Authors use the phrase sp for species. I think it would be more correct to add a point after sp (writing sp.) because it is an abbreviation.
Some of the paragraph titles could be improved as listed in the attached file. For example, for Line 169 I would suggest: the title of the paragraph could be improved writing as follows: Impacts of G15-treatment on pepper root rot, biological characteristics, and yield.
Furthermore, only for the sake of clarity, I suggest not to point out at the control as a "control treatment", but as a condition, because control is more a condition than a treatment. (Please, see the attached file)
Literature references are well structured and up-to-date, although I suggest for lines 259-265 to add any reference about the relation between ammonia N decreasing and root rot prevention, which seems to be a key role of G15 to prevent root rot? (Please, see the attached file)

Experimental design

As I listed in the attached file, I only suggest to spend a few words about your G15 fertilizers in Mat&Met too? You told about it in the Introduction, but I think a few words in the first paragraph of the Mat and Met section would fit.

Validity of the findings

No comment

Reviewer 2 ·

Basic reporting

The paper describes the effect of a biological fertilizer on pepper growth and the impact on microbial community and soil composition. In general it is consize, well written, and clearly structured. The introduction contains information of papers in Chines, which are hard to consult by non-Chinese people. More information regarding Bjerkandera would have been useful. The figures are of low quality. In contrast to what the paper describes, there is no comprehensive analysis of root rot resistance, only circumstantial data of the relative abundance of Fusarium (not absolute numbers), and an unclear way of disease monitoring. I added details in the end.

Experimental design

The topic of the paper is relevant because Biological control of plant disease is becoming more and more important. However, I have some concerns regarding the experimental set-up:
- Only 1 (pooled) soil sample per plot was analyzed for microbial community composition?
- The microbial community composition should have been analyzed before inoculation as well to be absolutely sure that changes in microbial community are due to inoculum
- Only 1 plant per plot was measured (see also remark below)? This is very strange. Was it a healthy or diseased plant? Considering the biological variation In plant growth, the number of plants analyzed is too limited
- The authors indicate that the control field was plowed and treated in the same way, but without adding the inoculum. However, the inoculum doesn’t only contain G15, but also contains pig manure, straw, urea,…
- Microbial community is examined at 1 time point only. At least the situation before inoculation should be included to see what the dynamics are due to the inoculum. In addition, how can we be sure that the starting situation of C and G are exactly the same?

Validity of the findings

The relevance of their findings could be discussed more clearly in the discussion. Furthermore, a lot of the differences on soil content (and hence microbial community) is probably due to organic compounds that are added to the microbial inoculum, rather than the microbial strain itself. This is not discussed at all.
Some unexpected results (low alpha diversity, low number of genera,…) are not mentioned in discussion. The discussion contains statements that are not backed up by data, e.g. L222 (number of pathogenic bacteria is not analyzed), L236 (disease resistance mechanism cannot be explored with microbial community analysis), biological agent G15 refers to the microorganisms, but this is applied together with organic compounds (manure,….) and therefore not possible to point to the microorganism as the major driver of these differences.

Additional comments

Other general and minor remarks
General remarks:
- Yield increases but also looked at damage? Maybe more PGP effect?
- A lot of references to papers (and patent? Hard to look up the content of this patent) in Chinese: not the best suited because a large part of the scientific community cannot consult these papers. In most cases these references can be changed by alternatives or excluded. It also seems that there are a lot of self-citation
- Check spelling and correct use of all family, genus, and species names, also in reference list
- Check citations and references. For instance, Fravel et al. 2005 is not in reference list
- Materials and methods could use more specific information for some crucial aspects (e.g. how many plants per plot, determination of disease index,…)
- Table legends should contain all information to interpret data, not a summary or conclusion of data

Abstract:
- Increased richness of beneficial bacteria is mentioned twice (L22, 27): combine


Introduction:
- Confusing start: Hymeophora subfamily? F. verticilliodes=F.moniliforme?;
- L37: “The pathogen invades”: specify “these fungal pathogens…”
- L47: not all of these references seem to be crucial, please avoid papers in Chinese for reason mentioned above
- L47: Ikeda 2010
- L53. Amyloliquefaciens; also in corresponding reference
- L55: revise
- L56: all references needed? Here I would expect a good review pointing to mode of action of BCOs


Materials and methods
- L85: how was the chemical composition assessed?
- L98: how many plants per plot of 300 m2? In table 1, it seems to be only 1 plant? Otherwise it would have been mentioned that the data in Table1 represents the averaged of X plants, and also a std dev would have been expected
- L102-103: “after selecting three batches”?? not clear
- How is plant height and root length measured? Authors provide data that are accurate to 0.01 cm?? I’m not sure if this is feasible.
- It is nor indicated how disease is monitored
- L112: stored
- L114: why sieving before DNA extraction? Could this create a bias? Why DNA extraction of entire soil sample?
- L131: specify length of DNA fragments extracted from gel
- It would have been better to do the Illumina analysis in duplicate (two technical repeats for each DNA sample)

Results:
- First title is not appropriate: the impact of root rot is not analyzed in detail.
- Table 1 could also include averages and stddevs of the 3 analyses per treatment; similar remarks for Tabe2
- L173-174: not clear how disease index or control effect is determined.
- L180: not described how interaction was assessed
- Table 3: 4 digits after comma? Sometimes 2 or 0 digits after comma? This should be cleaned up
- Quality of figures could be improved regarding resolution as well as font size/readability (in particular for fig 2 and 4)
- Unexpected that only 1 bacterial phylum was found?

---

## Round 0.2 · Minor Revisions

Please, follow all the remaining comments of the reviewer

·

Basic reporting

The article was written in clear, unambiguous, technically correct English and conforms to professional standards of courtesy and expression.
The Authors agreed with all of my suggestions and changed the paper accordingly. Probably just for material mistakes, a few of my previous suggestions were not fulfilled. That is the list:

Introduction
line 41. and chlamydospores to the (please, remove “the” as I suggested in my previous review, because to overwinter is a verb) overwinter in the….
Line 85. the grown plants, (please, add a comma before thus, not after it, as I suggested) thus maintaining

Mat & Met
Line 104 …… pig manure, straw,. (would you like to specify instead of using etc.?),
Line 104. ……….. , which (please, add “which” preceded by a comma) are necessary ……...
Line 107. ….. remove one of the “to”
Line 145. ……..water and fertilizer, (would you please, avoid etc. and specify thoroughly),

Results
Line 238. bacteria had an (please, remove n) richness of more….

Discussions.
Line 272, 275, 276. ……. the G15-treatment (I would leave G15 treated here, because “G treated” works as an adjective) plants
Line 285. ….. of the G15-treatment (I would leave G15 treated here because it works as an adjective) soils were ….

Experimental design

No comment.

Validity of the findings

No comment

---

## Round 0.3 · accepted · Accept

Thank you for your submission to PeerJ.

I am writing to inform you that your manuscript - Pepper root rot resistance and pepper yield are enhanced through biological agent G15 soil amelioration - has been Accepted for publication. Congratulations!